# Emotions, perceived threat, prejudice, and attitudes towards helping Ukrainian, Syrian, and Somali asylum seekers

**Sharon Xuereb**[ID]*

School of Psychology, Faculty of Arts and Social Sciences, The Open University, Milton Keynes, United Kingdom

* sharon.xuereb@open.ac.uk

## Abstract

Europe receives thousands of asylum seekers. This study examined whether European participants distinguish between European, Middle Eastern, and African asylum seekers in relation to positive and negative emotions, perceived threat, prejudice, and attitudes towards helping. The study also examined how these variables interrelate to influence each other. 287 participants were recruited from the UK and Malta. The study found that higher positive emotions and attitudes towards helping, and lower negative emotions, classical prejudice, and conditional prejudice were reported in relation to Ukrainian than Syrian or Somali asylum seekers. A mediation analysis was conducted to examine the interrelationship of these variables. Emotions and perceived threat had an indirect effect on attitudes towards helping, via prejudice. The effect of perceived threat was stronger than that of emotions. The study suggests that to improve people's openness towards helping asylum seekers, it is necessary to reduce the perception that asylum seekers are threatening.

## Introduction

The Geneva Convention [1] defines a refugee as someone who is unable to be protected in their own country, or fearful of not being offered such protection. Asylum seekers are refugees whose request for sanctuary is still being processed. According to the United Nations High Commission for Refugees [2], as of December 2020, there were 12 million refugees, internally displaced persons, asylum seekers, or stateless people living in Europe. Since 2014, large numbers of people have sought to come from Syria, Afghanistan and Iraq, with around 110,000 asylum seekers estimated to have arrived via the Mediterranean Sea in the first eight months of 2022 [3]. Since 2022, the Russian war on Ukraine has displaced millions more people and, as of October 2022, 7.7 million Ukrainians were seeking refuge across Europe [4].

Governments deal with asylum seekers from different countries in different ways. For example, in April 2022, the UK government started offering visas to people who travelled from Ukraine and paying British citizens to host Ukrainians in their own homes [5 https://www.gov.uk/register-interest-homes-ukraine]. However, during the same period, the UK government published a plan by which some asylum seekers who reach the UK via the English

**Data Availability Statement:** All relevant data are within the manuscript and its Supporting information files.

**Funding:** The author received no specific funding for this work.

**Competing interests:** The author has declared that no competing interests exist.

Channel would be sent on to Rwanda, for their claim of refuge to be processed there [6]. Denmark is considering a similar plan [7] and is also revoking residency to some Syrian asylum seekers, claiming that areas around Damascus are now safe to return to [8].

In general, people tend to overestimate the size of minority populations in their own country, while underestimating the size of the majority population [9]. The media and some politicians suggest that countries are being overrun by asylum seekers [10]. This increases the negative emotions that citizens feel towards asylum seekers or immigrants, including perceptions of threat [11, 12]. Citizens do not necessarily apply UNHCR definitions in their evaluations of asylum seekers, and they come to their own conclusions about who should be helped and under which circumstances [13], judging whole groups of people in terms of worth and moral performance [14].

There is agreement in the current research literature that perceived threat influences cognitive evaluations, such as prejudice, and negative emotions towards people [15, 16]. In addition, empirical studies have demonstrated that prejudice and negative emotions increase restrictive attitudes towards refugees and reluctance to help them [17, 18]. However there is a gap in the literature concerning how emotions, perceived threat, and prejudice may come together to influence attitudes towards helping different groups of asylum seekers. Research has shown that refugees, asylum seekers, undocumented migrants or other incomers who are Muslim or not White are seen as a drain on a country, in comparison to generally-White migrants (such as Germans, Canadians or British) who are perceived as assets [19]. However, it is still unclear whether European citizens distinguish between asylum seekers from Europe and those from other parts of the world. This paper presents a study of Europeans' perceptions of asylum seekers from Ukraine, Syria, and Somalia. The research investigated possible differences in the negative emotions, perceived threat, prejudice and attitudes that Europeans hold towards people arriving from each of these countries. In addition, the theoretical understanding of the relationship between these variables was statistically examined, to explore how they fit together and inform each other.

## Social identity theory

The way we categorise ourselves and others as 'groups' of people impacts our perceptions of ourselves and others. Social identity theory [SIT; 20, 21] proposes that individuals categorise themselves as belonging to specific groups, and that they identify their ingroup as distinct from an outgroup, i.e. individuals outside their group. Individuals evaluate their ingroup more favourably than an outgroup, and this in turn helps develop a more positive social identity for the individual [21]. In addition, unconsciously emphasising differences between the ingroup and outgroups [21] confirms our role within the social world [22]. If such perceptions operate in relation to a European ingroup, the theory would predict that asylum seekers from Ukraine, a European country, would be perceived more favourably than those from war-torn countries outside Europe, with the consequence that European citizens would be less willing to help the latter.

## Perceived threat

Intergroup perceived threat is experienced when one group considers that another group may cause them harm [23]. The supposed distinctiveness of an outgroup increases the likelihood that the group will be perceived as a threat [24]. There are realistic threats, such as threats to the ingroup's power, resources, or general well-being; and symbolic threats, including threats to the group's values, way of life, or belief system. Perceptions of threat, regardless of how

realistic or likely they are, have real consequences on perception of outgroup and on inter-group behaviour [25].

Numerous studies have found that asylum seekers are perceived as a threat, for instance, being seen as immoral individuals who are trying to con the system, or as harbouring infectious diseases [e.g. 26, 27]. Furthermore, asylum seekers are seen to pose greater economic, physical, cultural and social threats than other immigrants [28]. Host populations may believe there are finite resources, and that the population will lose some of these scarce resources to asylum seekers. Such thinking has an effect on prejudice, which in turn has an effect on attitudes towards accepting refugees [29]. For example, Turkish participants who were asked about Syrian refugees disclosed that the participants perceived Syrians as threatening control over Turkish territory. The perception led to negative emotions, which in turn led to lack of tolerance towards the Syrian refugees [30].

Perceiving immigrants or refugees as threatening leads to more negative attitudes towards them [16], with an even stronger effect than other personality-based variables such as right-wing authoritarianism [15]. Previous research has identified certain conditions which may increase perceived threat. For instance, highlighting differences between host populations and immigrants has an effect on how threatening the immigrants are perceived to be, and how much prejudice is shown towards them [31]. Moreover, immigrants from more distant cultures are perceived as more threatening [32]. Europeans prefer migrants and refugees who are of the same ethnicity and from European countries, as compared to those from a different ethnicity and from non-European countries [33, 34]. Furthermore, groups associated with a different religion are perceived as particularly threatening [35]; for example, Americans see Arabs and Muslims as less human than Americans or Europeans [by Americans; 36]. Hence, in Europe, one would expect refugees from another European country, like Ukraine, or those of Christian heritage to be perceived more favourably than refugees from the Middle East or Africa or of Muslim heritage, such as those from Syria or Somalia. In addition, there is evidence that ethnic groups from lower latitudes (closer to the equator) are perceived as less trustworthy, competent, and capable, when compared to people from higher latitudes [37], which also suggests that refugees from South of Europe would be perceived more negatively.

Another issue that influences perceptions is the current status of asylum seekers. In the European context, people from Ukraine can be perceived as resettled refugees, because European governments, such as the UK, issue visas for them before they start travelling, whereas people from other countries generally need to somehow make their way to Europe and only then apply for refugee status. Research in Australia [17] has compared attitudes towards resettled refugees and not-yet resettled asylum seekers. Participants reported significantly more anger and fear towards the asylum seekers, compared to resettled refugees. Arguably the non-settled ones may be seen as 'bogus' asylum seekers, or people who are trying to take advantage of the system. These findings are of major concern because [23] perceived threat may lead to a number of serious consequences, including cognitive biases in intergroup perceptions and emotional responses (such as hatred or righteous indignation), and the unwillingness to see the outgroup as fully human.

## Positive and negative emotions

Perceived threat leads to increased negative emotions towards refugees or asylum seekers, including anger, resentment, and anxiety [38, 39]. For example, research conducted in Germany indicated that concerns about cultural differences, financial strain, criminal acts, and conflicts within society led to negative emotions that included anger, fear, disgust and sadness [16]. However such negative emotions appear to be situation-dependent, at least to some

extent. That is, people feel anger when they think incomers are exploiting the system, whereas they feel sympathy towards people whom they perceive are genuine refugees who have gone through terrible experiences [40, 41]. Hence the public's perception of the legitimacy of asylum seekers influences feeling towards refugees.

Emotions towards asylum seekers in turn predict behaviour or behavioural attitudes towards them, which may have a serious impact on their well-being. Emotions predict either approach tendencies, such as trying to help, or aggressive tendencies [38]. Negative emotions function to legitimise or increase antisocial behaviour towards asylum seekers [42] and anger, fear, disgust, contempt, and lack of admiration towards refugees all translate into less favourable attitudes towards them, more support for harsher policies, and lower pro-immigration attitudes [17, 27, 43]. Furthermore, intergroup anxiety has been shown to lead to prejudice [18], indicating that this can be increased by both negative emotions and perceptions of the outgroup as perhaps threatening or worrying.

## Prejudice

Stereotypes are underlying beliefs about a group of people, and attributions of specific characteristic to them; they can be positive or negative [44, 45]. Some research has found that stereotypes about immigrants tend to be ambivalent, for example low warmth and high competence, or high warmth and low competence [46]. Refugees as a category are generally perceived to be low on both warmth and competence, though the background of the refugees has some impact on perception of warmth [47]. Stereotypes can lead to prejudice, and are therefore key to our understanding of this phenomenon.

Prejudice tends to be a devaluation of people because of their group membership, leading to potential harm or negative consequences [48]. It relates to a judgement made before, or irrespective of, actually meeting the person. For instance, dark-skinned young people in Australia have been perceived as religiously-indoctrinated terrorists simply because of their appearance [44, 49]. Prejudice is a set of hostile attitudes or feelings towards someone [44], following an evaluation of the group the person belongs to [45]. It is both irrational, in that prejudice persists in the face of evidence to the contrary, and negative [44]. Prejudice can be seen as an evaluation that includes both cognitive and affective responses [29], and often reflects a perception of the other person as low on warmth and competence [50].

There is limited empirical evidence examining people's perceptions of refugees from different ethnicities or cultures. With regard to migration, migrants are perceived as more or less desirable depending on their country of origin [51]. Specifically, UK-based people oppose migrants who are not white and who are from culturally more distinct origins more than they oppose white immigrants from culturally more similar origins. In that, historically, British and European citizens are of a white ethnicity, one could link this to ethnocentrism, that is the belief that group differences are real and important, and that one's ethnic group is superior to other groups [52], and which has an effect on prejudice [53].

More recently, a distinction has been made between classical (blatant, explicit, overt) and modern (conditional, subtle, indirect) prejudice [54, 55]. Classical prejudice refers to blatantly negative statements about a group of people whereas conditional prejudice is more subtle. An example of conditional prejudice would be when someone suggests that they do not have a problem with migrants or refugees but are simply concerned about their well-being or how they would fit in the host population. This is a more socially acceptable type of prejudice, and perhaps one that is more common in media or political discourses.

Prejudice may lead to abuse, avoidance, discrimination, or violence against the person. It is argued that prejudice does have some automatic aspects, though it is not fully automatic and

can be controlled by the prejudiced individual [50]. Ingroup norms about prejudice impact the amount of prejudice shown, with prejudice more likely when ingroup norms support it [41]. This would suggest that discourses that are prejudicial about refugees, particularly if refugees are from ethnicities that are perceived as less valuable, will facilitate further prejudice. This is a serious issue, as, for both resettled refugees and asylum seekers, prejudice predicts more restrictive social policy attitudes [17].

### Attitudes towards helping

Social justice can only be achieved when everyone, irrespective of their immigration status, has access the economic resources they need, and there is no disadvantage in the distribution of resources [56]. Support is therefore a key element of social justice. As has been noted, most populations make their own assessments of the support and help asylum seekers deserve, regardless of asylum seekers' legal status [57]. Therefore, exploring predictors of how help should be apportioned is an important area of study, particularly as many asylum seekers have additional needs for support, for instance, because of mental health problems such as anxiety, depression, and post-traumatic stress disorder, and experiences of sexual violence during transit [58, 59]. There has been little research related to help for asylum seekers although a study using a Turkish sample focused specifically on behavioural intentions towards Syrian refugees [60]. Participants were asked whether they would befriend a Syrian refugee, or sign a petition to support them. The research found that perception of threat was a strong predictor of behavioural intentions.

### The current study

There is empirical evidence that perceived threat, emotions, and prejudice have an effect on attitudes towards helping refugees or asylum seekers. The inter-relationship between these factors has been measured in relation to outgroups. For example, perceived threat predicts emotions and behavioural intentions, and emotions predict behaviour intentions [61]. However participants in that study were not asked to consider asylum seekers. In addition, it is argued that different outgroups are perceived as posing different types of threat, also depending on the ingroups' context [62]. While there are indications that European participants would look more favourably upon Ukrainian asylum seekers, as compared to Middle Eastern or African ones, empirical evidence is currently lacking.

This study addressed the following research questions:

1. Is there a significant difference in how Ukrainian, Syrian, and Somali asylum seekers are scored on perceived threat, emotions, prejudice, and attitudes towards helping?

2. To what extent does prejudice mediate the relationship between emotions and perceived threat, and attitudes towards helping?

## Materials and methods

### Design

The study was a between-participants design. Participants from Malta and the UK were asked to focus on Ukrainian, Syrian, or Somali refugees while completing a survey. The selection of the UK and Malta, and the three countries of origin was based on the following rationale:

a. Ukrainian asylum seekers featured regularly in the news at the time the study was conducted. Ukrainians are European and of Christian heritage.

b. Syrian asylum-seekers have also featured in the news in recent years. Syrians are Middle Eastern, not European, and are of Muslim heritage. Syria is geographically closer to Malta than Ukraine and there are some overlaps between Maltese and Syrian culture.

c. Somali asylum seekers have featured little in the news in recent years. Somalis are African, not European, and of Muslim heritage.

The UK was chosen because it is the researcher's country of residence. Malta was chosen because it receives large numbers of migrants from the Mediterranean, and most Maltese people have met or regularly interact with asylum seekers. In 2021, 1.8% of the Maltese population comprised refugees [63], although this understates the number of asylum seekers who live in Malta, as, for the same year, only 8% of asylum seekers were granted refugee or subsidiary protection status [64].

## Participants

Data was collected between June and July 2022. 303 people participated in a questionnaire study. Of these, 10 did not complete an entire scale or more, and were therefore removed. Another 6 were deemed to be multivariate outliers (see data screening explanation further down), and were also removed. Therefore the final sample comprised 287 participants. This number is appropriate for a mediation analysis (minimum sample 118) [65] and an ANOVA with 3 levels for Ukrainian, Syrian and Somali (a sample of 252 needed for a medium effect: G*Power) [66].

The sample included 95 males, 189 females, and 2 of non-binary/third gender. 130 participants lived in the UK and 157 lived in Malta. Of the participants, 266 were born in the country they resided in, 20 were born in a different country, and 1 did not answer this question. Table 1 shows a breakdown of participants' ages.

## Materials

Participants were asked four initial questions, about their age, gender, country of residence (only those residing in Malta or the UK could continue), and whether they had been born in the country they resided in. They then completed the four scales detailed below. No information that could identify participants was collected.

**Emotions.** Participants were asked how they feel about asylum seekers, on a scale from 1 to 7, from 'not at all' to 'extremely'. The emotions were identical to those used in previous studies about asylum seekers [27, 67]: admiration, fondness, inspiration, pride, respect, anger,

**Table 1. Ages of participants.**

|  |  | N | Age (Mean) | Age (SD) |
|---|---|---|---|---|
| **UK** | Males | 30 | 52.93 | 15.44 |
|  | Females | 83 | 51.89 | 14.15 |
|  | Non-binary/3rd gender | 1 | 40.00 | - |
|  | Total UK | 114 | 52.06 | 14.42 |
| **Malta** | Males | 57 | 53.33 | 13.99 |
|  | Females | 83 | 47.23 | 13.52 |
|  | Non-binary/3rd gender | 1 | 76.00 | - |
|  | Total Malta | 141 | 49.93 | 14.11 |
| **Total** |  | 255 | 50.88 | 14.26 |

32 participants did not give their age

shame, contempt, disgust, frustration, hate, resent, unease, pity, and sympathy. Both previous studies had identified a single factor they called 'contempt', comprising the 8 negative emotions. Some previous researchers identified a factor 'lack of admiration' and developed 'pity' as an average of the variables pity and sympathy [27]. Others used 'lack of admiration', and dropped pity and sympathy [41].

In the current study, a principal components factor analysis was run, to examine the factor structure of the emotions scale [67]. As in previous studies, there was a Negative emotions factor comprising disgust, resent, anger, shame, hate, frustration, unease, and contempt. The remaining positive emotions variables loaded on the same factor, that is fondness, pride, inspiration, admiration, sympathy, respect, and pity. Hence this two-factor structure was used. Negative emotions had a Cronbach's alpha of .85, and positive emotions of .91. The term 'positive emotions' was used instead of Warmth, in order to distinguish from Warmth as described in studies about warmth and competence [50].

**Perceived threat.**   This scale was based on a previous analysis of realistic and symbolic threat [68], and on a measure for perceived threat of immigrants by Italians [69]. The scale comprised 3 items for Realistic threat (e.g. 'I feel threatened'), and 3 items for Symbolic threat (e.g. 'I think the economical resources gained by them will likely damage the people of my country'). Items were rated on a 7-point Likert scale, from 'not at all' to 'extremely'. In the current study, a principal components factor analysis showed that the 6 items were best specified in a single factor, and therefore a single factor of perceived threat was used. Cronbach's alpha was .87.

**Prejudice.**   The Prejudice Against Asylum Seekers (PAAS) scale [55] was used to measure prejudice. This comprises two sub-scales: classical prejudice (8 items, e.g. 'They are not welcome in our country') and conditional prejudice (8 items, e.g. 'They should return to their country once safe to do so'). A principal components factor analysis showed that 2 items determined as part of conditional prejudice instead loaded on classical prejudice. These were: 'They need help, however there are people in our country who need the help more' and 'They might struggle to integrate into our system'. These items were kept in the conditional prejudice subscale because they had the lowest loading on classical prejudice, and this could have been a sample-specific finding. Internal reliability was as follows: Classical prejudice .93, and Conditional prejudice .84.

**Attitudes towards helping.**   Finally, participants were asked to what extent they thought asylum seekers should receive certain types of help, such as medical issues or updating their qualifications. They rated their agreement with the statements on a 7-point Likert scale, from 'not at all' to 'extremely'. A principal components factor analysis showed that the data was best explained by a single factor. Cronbach was .94.

## Procedures

Ethical approved was obtained from The Open University Human Research Ethics Committee (reference 4398). Advertisements were posted on a number of British and Maltese social medial groups, and members were encouraged to share the advertisement. Data collection took place between June and September 2022. Interested people could click on the link in the advert, which took them to the Qualtrics survey. On the anonymous Qualtrics questionnaire, participants were first presented with information about the study, and where then asked to mark their consent in relation to key elements (e.g. that they were over 18 years of age). Participants marked their consent on the online survey, and those who selected 'no' to any of the consenting questions were unable to complete the survey, and were simply thanked for their time. Demographic questions were presented first. The ensuing four scales were presented in a

randomised order. Within each scale, questions were also randomised in their presentation. Using Qualtrics (Qualtrics, Provo, UT), participants were randomly assigned to a Ukrainian, Syrian, or Somali refugee condition. With regard to the 287 completed questionnaires, 105 (36.6%) focused on Ukrainian asylum seekers, 92 (32.1%) on Syrian asylum seekers, and 90 (31.4%) on Somali asylum seekers. Participants then answered the same four scales.

## Results

Descriptive statistics were run for the total sample, broken down by gender, by country of origin, and by migration status. These are presented in Table 2.

Oneway ANOVAs were run to examine whether the gender and country differences were significant. With regard to gender, the two cases that reported not being male or female were excluded from the analyses. The only significant difference found was for classical prejudice, with women scoring lower than men, though the effect size was quite small according to guidelines [70]: $F(2,284) = 4.07$, $p < .05$, $\eta^2 = .03$.

With regard to migration status, this was a small group of 20 participants. They scored significantly lower than participants born in the country they resided in on conditional prejudice ($F(1,284) = 14.99$, $p < .001$, $\eta^2 = .05$), and the scores for attitudes towards helping approach significance with migrants reporting more positive attitudes towards helping ($F(1,284) = 3.77$, $p = .053$, $\eta^2 = .01$), but no other scores were significantly different between the two groups.

A Factorial ANOVA was run to examine whether Ukrainian, Syrian, or Somali asylum seekers were perceived differently. This was done for the total sample, and also split by country. While Malta and the UK are both European countries of Christian heritage, similar to Ukraine; Malta is geographically and culturally closer to Syria than is the UK, despite the religious differences. Ethnically and religious-culturally, Somalia is different to both the UK and Malta, though Somalis are well-represented amongst the asylum seeking population in Malta (9% of refugee applications in 2021) [64], which is something most Maltese are aware of. Table 3 presents descriptive statistics.

In terms of participants' country (see Table 2), there were significant main effects for negative emotions ($F(1,281) = 6.05$, $p < .05$, $\eta_p^2 = .02$) and perceived threat ($F(1,281) = 4.60$, $p < .05$, $\eta_p^2 = .02$). That is, participants in Malta reported higher ratings on negative emotions and conditional prejudice, and lower ratings on positive emotions and attitudes towards helping, than participants in the UK. With regard to asylum seekers' country of origin, there were significant main effects for negative emotions ($F(2,281) = 14.05$, $p < .001$, $\eta_p^2 = .09$), positive emotions ($F(2,281) = 20.39$, $p < .001$, $\eta_p^2 = .13$), conditional prejudice ($F(2,281) = 3.87$, $p < .05$, $\eta_p^2 = .03$), and attitudes towards helping ($F(2,281) = 3.295$, $p < .05$, $\eta_p^2 = .02$). None of the interaction effects were significant.

**Table 2. Means and standard deviations (in brackets) of key variables.**

|  | Males (n = 96) | Females (n = 189) | UK (n = 130) | Malta (n = 157) | Born in country (n = 266) | Migrant (n = 20) | Total (N = 287) |
|---|---|---|---|---|---|---|---|
| **Negative emotions** | 12.07 (5.37) | 10.89 (4.27) | 10.50 (4.36) | 11.92 (4.86) | 11.35 (4.79) | 10.45 (2.98) | 11.28 (4.68) |
| **Positive emotions** | 28.31 (9.51) | 30.17 (9.05) | 29.30 (10.37) | 29.77 (8.24) | 29.28 (9.21) | 33.15 (9.48) | 29.56 (9.25) |
| **Perceived threat** | 13.20 (7.05) | 12.59 (6.42) | 11.85 (6.27) | 13.57 (6.80) | 13.00 (6.67) | 10.30 (5.31) | 12.79 (6.61) |
| **Classical prejudice** | 19.93 (10.95) | 16.29 (9.97) | 15.48 (10.47) | 19.24 (10.18) | 17.81 (10.55) | 13.56 (8.54) | 17.54 (10.46) |
| **Conditional prejudice** | 37.33 (8.34) | 35.03 (11.50) | 33.21 (12.09) | 27.88 (8.83) | 36.42 (10.25) | 27.05 (12.71) | 35.76 (10.67) |
| **Attitudes towards helping** | 51.77 (11.65) | 52.72 (12.22) | 52.25 (12.93) | 52.60 (11.19) | 52.10 (12.04) | 57.47 (10.33) | 52.44 (11.99) |

Notes: Two participants reported being of a 3rd or non-binary gender, and were not included in the gender split. One participant did not answer the question re whether they were born in the country they resided in.

**Table 3. Means and standard deviations for variables, split by asylum seeker and participant country of origin.**

|  | Ukraine | | | Syria | | | Somalia | | |
|---|---|---|---|---|---|---|---|---|---|
|  | UK (n = 50) | Malta (n = 55) | Total (n = 105) | UK (n = 42) | Malta (n = 50) | Total (n = 92) | UK (n = 38) | Malta (n = 52) | Total (n = 90) |
| Negative emotions | 8.90 (1.74) | 10.09 (3.29) | 9.53 (2.72) | 10.64 (3.50) | 12.57 (4.02) | 11.69 (3.89) | 12.45 (6.42) | 13.24 (6.27) | 12.90 (6.31) |
| Positive emotions | 33.74(10.31) | 33.51 (8.43) | 33.62 (9.33) | 28.93 (8.91) | 27.62 (7.11) | 28.22 (7.97) | 23.87 (9.45) | 27.88 (7.78) | 26.19 (8.71) |
| Perceived threat | 11.12 (5.13) | 12.44 (6.54) | 11.81 (5.92) | 11.95 (5.66) | 14.62 (7.04) | 13.40 (6.55) | 12.71 (8.07) | 13.75 (6.79) | 13.31 (7.33) |
| Classical prejudice | 11.70 (6.69) | 14.71 (8.27) | 13.28 (7.67) | 15.86 (9.43) | 21.01 (10.44) | 18.66 (10.27) | 20.05(15.54) | 22.33 (10.23) | 21.37 (11.72) |
| Conditional prejudice | 30.44(10.52) | 36.20 (8.64) | 33.46 (9.96) | 35.60(12.12) | 38.49 (9.49) | 37.17 (10.81) | 34.21(13.50) | 39.08 (8.25) | 37.02 (10.98) |
| Attitudes towards helping | 55.12(12.05) | 53.48 (11.95) | 54.26 (11.97) | 52.31(12.11) | 52.89 (10.46) | 52.62(11.18) | 48.39(14.21) | 51.38 (11.15) | 50.12 (12.54) |

Tukey HSD post-hoc tests were run to further understand the main effects of asylum seekers' country of origin. These indicated that, for the variables where there was a significant effect, Ukrainian asylum seekers were rated more positively than Syrians, and also than Somalis. The significance levels for the total sample were as follows: negative emotions Ukraine-Syria p = .01, negative emotions Ukraine-Somalia p < .001, positive emotions Ukraine-Syria p < .001, positive emotions Ukraine-Somalia p < .001, classical prejudice Ukraine-Syria p < .001, classical prejudice Ukraine-Somalia p < .001, conditional prejudice Ukraine-Syria p < .05, and conditional prejudice Ukraine-Somalia p = .051. For attitudes towards helping, where the main effect was close to the .05 threshold, Ukrainians were rated more positively than Somalis (p < .05).

In preparation for running the mediation analysis, confirmatory factor analyses were run, using AMOS (Version 27). In preparation, correlations between the variables were examined. As shown in Table 4, classical prejudice correlated highly with all the other variables. There were also high correlations between negative emotions and perceived threat, and positive emotions and attitudes towards helping.

AMOS did not support a confirmatory factor analysis with all the variables, as a positive definite solution was not found. This was possibly due to the high correlations between some of the variables, even though multicollinearity had been addressed earlier in the data screening process. Hence two analyses were run, with negative emotions and positive emotions separately. After acknowledging the covariance of some error terms, both models had a reasonable fit: negative emotions: $\chi^2$(263)624.17 p < .001; PCMIN = 2.37, GFI = .86, CFI = .93, TLI = .91, RMSEA = .07(.06-.08), SRMR = .06; positive emotions: $\chi^2$(252)676.54 p < .001; PCMIN = 2.69, GFI = .84, CFI = .93. TLI = .91, RMSEA = .08(.07-.08), SRMR = .07. The statistics suggesting issues with the model were $\chi^2$, PCMIN, and RMSEA, which are the ones that suffer most bias with larger samples such as in the current study.

**Table 4. Pearson's correlations for variables (N = 287).**

|  | Negative emotions | Positive emotions | Perceived threat | Classical prejudice | Conditional prejudice | Attitudes towards helping |
|---|---|---|---|---|---|---|
| Negative emotions | 1 | -.36 | .64 | .66 | .33 | -.42 |
| Positive emotions | -.36 | 1 | -.33 | -.61 | -.52 | .68 |
| Perceived threat |  |  | 1 | .69 | .49 | -.48 |
| Classical prejudice |  |  |  | 1 | .62 | -.66 |
| Conditional prejudice |  |  |  |  | 1 | -.57 |
| Attitudes towards helping |  |  |  |  |  | 1 |

Note: All correlation values significant at p < .001

Two mediation analyses were then run, for negative emotions (see Fig 1) and positive emotions (see Fig 2) separately. Both models had a reasonable fit: negative emotions: $\chi^2(263)624.17$ p < .001, PCMIN = 2.37, GFI = .86, CFI = .93, TLI = .91, RMSEA = .07(.06-.08), SRMR = .06; positive emotions: $\chi^2(254)693.30$ p < .001, PCMIN = 2.73, GFI = .84, CFI = .92, TLI = .91, RMSEA = .08(.07-.09), SRMR = .08.

For the negative emotions model, the indirect effect of perceived threat on attitudes towards helping was β = -.76, p < .001, and the indirect effect of negative emotions on attitudes towards helping was β = -.33, p < .001. Hence negative emotions was a significant mediator variable in these relationships, with the indirect effects being larger than the direct effects.

For the positive emotions model, the indirect effect of perceived threat on attitudes towards helping was β = -.72, p < .001, and the indirect effect of negative emotions on attitudes to help was β = .26, p < .001. Therefore prejudice was a mediator variable for both relationships, as the indirect effects were larger and more significant than the direct effects.

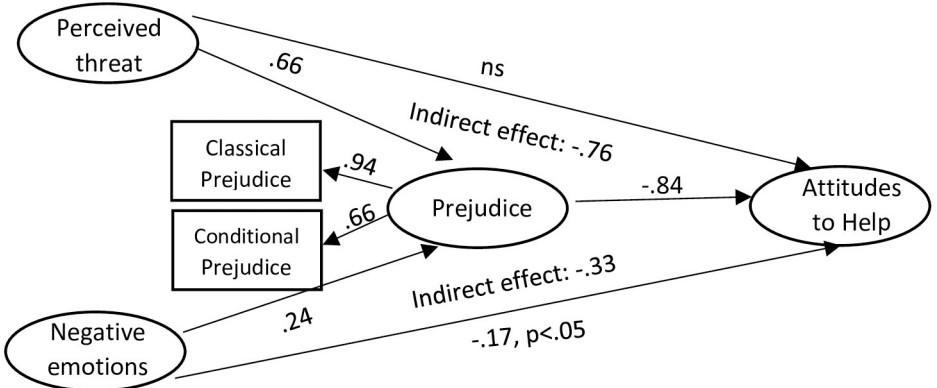

**Fig 1. Mediation path analysis for negative emotions.** Note: Unless otherwise stated, all standardized regression weights (β) valid at p < .001.

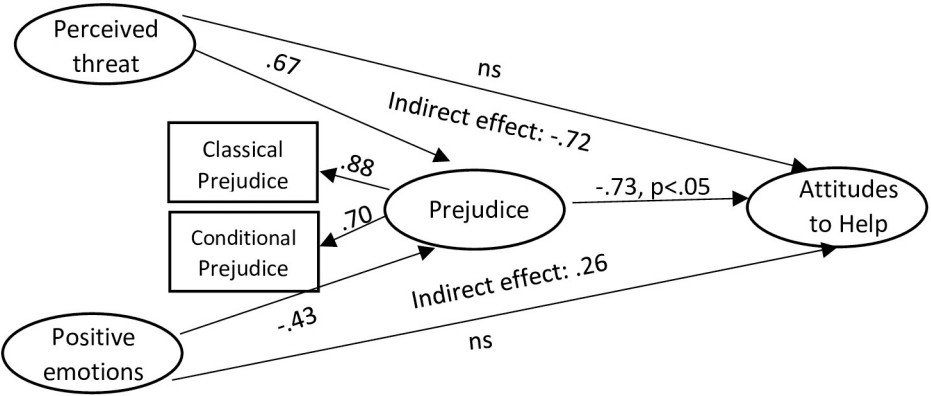

**Fig 2. Mediation path analysis for positive emotions.** Note: Unless otherwise stated, all standardized regression weights (β) valid at p < .001.

## Discussion

In this study, 287 participants from the UK and Malta answered questions about perceived threat, prejudice, emotions, and attitudes towards helping asylum seekers from Ukraine, Syria, and Somalia. Across the entire sample, more negativity was reported about Syrian and Somali asylum seekers than Ukrainians, with more negative emotion, less positive emotion, more prejudice, and lower attitudes towards helping being disclosed. Prejudice mediated the relationship between perceived threat and emotions, and attitudes towards helping; meaning that, perceived threat and negative emotions led to higher prejudice, which in turn led to lower attitudes towards helping, while positive emotions led to lower prejudice, which then led to higher attitudes towards helping.

Maltese participants reported more negative emotions, perceived threat, and prejudice than participants from the UK. Across both the UK and Malta, two groupings of the three countries of origin (Ukraine, Syria, and Somalia) emerged consistently: Ukrainians received a more positive response, whereas Syrians and Somalis were perceived similarly, and less favourably.

UK adults are less positive about both economic migrants and asylum seekers who are not white and who are perceived as culturally more distant [47]. The results show that not all asylum seekers are equal, and White European asylum seekers are seen in a more positive or kinder light than darker-skinned ones from outside Europe. Even though Syria, with its Mediterranean culture, can be seen more similar to Malta, Maltese participants expressed more negative perceptions of Syrians than of Ukrainians. This is despite emotional scenes on the media in the last few years, such as images of Africans in boats in the Mediterranean Sea, or the death of Syrian children as they tried to make it to Europe. This suggests that empathic responses towards asylum seekers may be present for a period of time, but perhaps fatigue sets in and perceptions become less positive. More research is needed to explore this process of change, and reasons for it.

Considered in terms of Social Identity Theory [21], the results suggest that a White European identity and possibly an identity of Christian heritage become salient when Europeans consider asylum seekers. If Syrians and Somalis but not Ukrainians are positioned as members of outgroups, this would explain the more negative perceptions of non-European asylum seekers. Hence Europeans do not just prefer Europeans as economic migrants [33], but also as asylum seekers. It is possible that the less familiar cultures, as well as the difference in religious heritage, are seen as more threatening [32, 35]. Syrians' and Somalis' distinctiveness as an outgroup, whether in religion, dress, or other cultural behaviours, means they may be seen as more threatening [24].

The literature about intergroup threat identifies two types of threat: realistic and symbolic [25]. The confirmatory factor analysis in the current study questions this distinction. Looking at media and political discourses, politicians may emphasise symbolic threat, with media focussing more on realistic threats [71]. Possibly the various discourses lead people to experience threat more 'holistically', rather than in its theoretical subtypes. Further study could examine how people experience these theoretical sub-types of threat.

In line with previous studies [23], this study found that perceived threat leads to prejudice towards asylum seekers. The finding that both emotions and perceived threat have an effect on prejudice confirms the understanding of prejudice as having both an affective and a cognitive element [29]. Furthermore, previous studies have identified important direct relationships between prejudice and attitudes, including that both prejudice and emotions (fear and anger) predict negative social policy attitudes about asylum seekers and resettled refugees [17, 18]. This study confirmed the role of prejudice as a mediator variable in this two-step relationship. Recognising that people evaluate asylum seekers as more or less deserving of help [57], we can

now understand further how these assessments are made. While perceived threat does influence intentions to help [60], and is associated with more negative attitudes [15, 31], this involves a further evaluation of asylum seekers. Hence there is a two-step evaluation: level of threat, and negative attitudes (prejudice). The effect of perceived threat on prejudice suggests that people who do experience prejudice should not simply be vilified as 'bigots' or such. They should be engaged with, to develop a deeper understand of why they feel threatened. If perceptions of threat are addressed, the current study indicates that prejudice should reduce, and consequently attitudes towards helping become more positive.

In terms of negative emotions, known to predict unfavourable attitudes and support for harsher policies for refugees [17, 27], the impact on attitudes towards helping is achieved via the experience of prejudice. Therefore emotions, such as intergroup anxiety, impact prejudice towards asylum seekers [18], and this in turn leads to reduced attitudes towards helping. Both positive and negative emotions had a stronger relationship with classical prejudice than with conditional prejudice. With classical prejudice being more overt or blatant [54, 55], this suggests that conditional prejudice is possibly not simply a more acceptable way to articulate prejudicial evaluations, but perhaps a less strong or less negative bias. That is, the distinction might be more of experience than of expression.

The indirect effect of perceived threat on attitudes towards helping (via prejudice) was stronger than that of emotions on attitudes towards helping. This indicates a stronger cognitive than emotional, effect on attitudes towards helping. In other words, what people think about asylum seekers and possibly how they perceive them, and the implications of accepting them, has a stronger effect than their feelings about asylum seekers. However, unsurprisingly, it may be difficult to separate the different influences. There is agreement in the current research literature that perceived threat influences negative attitudes and emotions towards refugees [15, 16]. Perhaps because of indications that cognition and emotions interrelate [72], it is necessary to focus on how they work together to influence attitudes towards asylum seekers. This is an outstanding point of interest which requires further investigation.

In practice, prejudice may be more effectively reduced by addressing perceptions of threat, which could in turn improve attitudes towards helping asylum seekers. For example, the perceptions that darker-skinned people are potential terrorists [49] or that there are excessively large numbers of asylum seekers [9] can be addressed.

Therefore it appears that prejudice, as a set of hostile attitudes [45], does not pre-empt the experience of perceived threat, negative emotions, or lack of positive emotions; but it is these experiences that lead to prejudice, which in turn compromises people's openness to supporting help for asylum seekers. The perceived desirability of asylum seekers, influenced by their country of origin [51], may impact how citizens perceive them and feel about them, and possibly, underlying emotions around feeling threatened serve to justify negative attitudes towards asylum seekers, hence further leading to their embedding. Beliefs such as that asylum seekers would be safer in their countries of origin can subsequently serve to justify lack of willingness towards asylum seekers receiving help, confirming the negative impact of less blatant or overt prejudice [55]. These are important findings in relation to work being done to reduce prejudice, or increase willingness to help asylum seekers. They indicate that work needs to be done on two main fronts, i.e. support people challenge beliefs that asylum seekers are threatening, and, in parallel, help people empathise with asylum seekers, so they can experience and acknowledge positive emotions towards them.

This study measured attitudes towards helping by asking participants to what extent they thought asylum seekers should receive help for a list of issues. Therefore the examples of help were distanced from participants, in that they were not asked if they as individuals would be willing to offer these kinds of help. Even in such a scenario, prejudice was an obstacle to

supporting such help, indicating support for political discourse and decisions around absolving countries of responsibility towards asylum seekers. In the face of discourses presenting asylum seekers as a threat, and encouraging reduced positive emotions towards them [e.g. 9, 10], this indicates that current discourses may serve to propagate prejudice, and in turn, reinforce lack of willingness to offer help.

This study took place against a political context in which Ukrainian asylum seekers were offered more protection by European countries, during a period where processes for receiving asylum seekers from many African or Asian countries became harsher. As already noted, citizens experience more fear and anger against unsettled refugees than settled ones [17]. There is arguably a two-way interaction between political and media discourse. For example, politicians state that Ukrainians are deserving of refuge and this view is reported in the media. This may contribute indirectly to prejudice against Asian and African refugees and perceptions that they are threatening. Perceiving refugees as threatening predicts more negative attitudes about them [16], which would inform speech and behaviour. On the other hand, it may be the higher levels of negative emotions, prejudice, and perceived threat towards non-European refugees, as evidenced in this study, that gives permission to politicians to make different decisions regarding their treatment. Of concern is that, if this is the common perception towards many asylum seekers, positive change becomes very difficult. Indeed, being part of a group that is seen to support prejudice further leads to such prejudice [41]. In terms of practical implications, it is therefore imperative that decision-makers reflect on how their use of discourse facilitates further exclusion of vulnerable individuals. On the other hand, citizens must be presumed to have some agency and ability to distinguish and interpret discourses around asylum seekers. Discussions of bias or racism, whether in the media or in schools, should cover the current experiences of people seeking refuge, in order to nurture learning and change.

### Limitations and recommendations for further research

All studies have limitations. While this was a good-sized sample, it was also self-selecting, because participants responded to advertisements placed on a small number of social media groups. It is possible that people who are more ambivalent about asylum seekers, with less extreme views, are not represented in the sample. Furthermore, citizens who engage less with social media, such as the elderly, may be under-represented in this study.

The reasons for favourable or unfavourable attitudes were not examined, so it is unclear to what extent, for example, cultural heritage, religious heritage, or skin colour impacted participants' responses. This is something that can be explored in future studies, that is, seeking to understand how citizens develop perceptions of threat, and basis for prejudice, towards asylum seekers from a range of cultural backgrounds. As recent studies are doing [28], it would also be helpful to explore how people from host countries think refugees may bring benefits, delving into potential impact of asylum seekers' country of origin with regard to this.

Data were collected in the Summer of 2022, when the war in Ukraine was very much in the media, with Western countries rallying around the people of Ukraine, who were represented in the media as particularly vulnerable and experiencing suffering. It is unclear whether such perceptions will remain in a few years, when possibly people in the UK and Malta become more weary of hosting Ukrainian asylum seekers. It would therefore be insightful to conduct a longitudinal study exploring how perceptions of asylum seekers develop during a protracted period of difficulty in the country of origin, whether war, hunger, environmental crises, or other reasons; exploring how perceptions of threat, emotions, or attitudes towards helping may change.

The influence of the wider context, such as political decisions, mainstream media, and discussions on social media need to be examined further. Specifically, it is important to understand how people engage with such messages and make sense of them, and to what extent such messages fuel prejudice or perceived threat, and influence emotions about and attitudes towards helping asylum seekers.

Participants were asked to what extent asylum seekers should receive help with a range of issues. It would be insightful to explore people's rationale regarding why and under which conditions they think helping asylum seekers is appropriate. This would help us understand further how underlying perceptions impact attitudes towards helping.

Finally, in terms of the mediation analysis, evidencing mediation is challenging [73], and the mediation analysis in this study has its weaknesses. It is acknowledged that caution should be used when inferring mediation from cross-sectional data, as this type of data collection does not account for the temporal element of the experience of variables [74]. That is, a cross-section mediation analysis may yield different results when compared to a longitudinal study examining the same variables.

## Supporting information

**S1 Text.**
(SAV)

## Author Contributions

**Conceptualization:** Sharon Xuereb.

**Formal analysis:** Sharon Xuereb.

**Investigation:** Sharon Xuereb.

**Methodology:** Sharon Xuereb.

**Project administration:** Sharon Xuereb.

**Writing – original draft:** Sharon Xuereb.

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
