## [Decision Letter · Decision Letter 0]

11 Jul 2023

PONE-D-23-14284Emotions, Perceived Threat, Prejudice, and Attitudes to Help Ukrainian, Syrian, and Somali Asylum SeekersPLOS ONE

Dear Dr. Xuereb,

Thank you for submitting your manuscript to PLOS ONE. After careful consideration, we feel that it has merit but does not fully meet PLOS ONE’s publication criteria as it currently stands. Therefore, we invite you to submit a revised version of the manuscript that addresses the points raised during the review process.

Two Reviewers have suggested moderate revisions. I encourage the Author to provide modifications especially in terms of theoretical background and completeness in reporting results.==============================

We look forward to receiving your revised manuscript.

Kind regards,

Stefano Triberti, Ph.D.

Academic Editor

PLOS ONE

Journal Requirements:

Reviewers' comments:

Reviewer's Responses to Questions

**Comments to the Author**

1. Is the manuscript technically sound, and do the data support the conclusions?

Reviewer #1: Yes

Reviewer #2: Yes

2. Has the statistical analysis been performed appropriately and rigorously? 

Reviewer #1: Yes

Reviewer #2: Yes

3. Have the authors made all data underlying the findings in their manuscript fully available?

Reviewer #1: Yes

Reviewer #2: Yes

4. Is the manuscript presented in an intelligible fashion and written in standard English?

Reviewer #1: No

Reviewer #2: Yes

5. Review Comments to the Author

Reviewer #1: The study investigates how Europeans perceive asylum seekers from various regions (Ukraine, Syria, and Somalia) in relation to emotions, perceived threat, prejudice, and willingness to help. The study included 287 participants from the UK and Malta. The results showed that participants reported more warmth and willingness to help, and less contempt, classical prejudice, and conditional prejudice towards Ukrainian asylum seekers compared to Syrian or Somali asylum seekers. The study suggests that prejudice mediated the relationship between emotions (contempt and warmth), perceived threat, and attitudes to help. The study implies that reducing the perception of asylum seekers as threatening can improve people's willingness to help.

The research addresses a highly relevant and timely issue, shedding light on perceptions and attitudes towards asylum seekers from different regions. The choice to focus on asylum seekers from Ukraine, Syria, and Somalia allows for an interesting comparative analysis.

Prejudice and stereotypes are two distinct but interrelated concepts in the field of social psychology. The distinction between these concepts might be clouded by the fact that social psychologists have not always been consistent in their use of these terms, and the definitions might slightly vary across different theoretical perspectives or cultural contexts. Prejudice refers to an unfavorable attitude, emotion, or action towards members of a certain group solely based on their membership in that group. It is important to note that prejudice is typically characterized by its affective or evaluative component (i.e., it is about feelings or evaluations about a group). On the other hand, stereotypes are generalized beliefs or expectations about the characteristics of members of a group. They are cognitive structures that contain the perceiver's knowledge, beliefs, and expectations about a human group. Stereotypes can lead to prejudice, and prejudice can reinforce stereotypes. For example, a person might hold a stereotype that members of a certain group are lazy (a cognitive belief), and because of this stereotype, they may develop a prejudice against that group (an affective response). However, it is important to understand that one can hold a stereotype about a group without necessarily being prejudiced against that group. For instance, someone might believe in the stereotype that all Asians are good at math, but this does not necessarily mean they harbor negative feelings towards Asians (prejudice). It seems to me that in some places of the text, these distinctions are not made clearly enough. In addition, the label "Warmth" can be confused with the content of a stereotype (warmth and competence in the stereotype content model). So I would suggest changing "Warmth" to "Warm". Alternatively, the paper could just use labels "negative emotions" and "positive emotions".

"However there is a gap in the literature in understanding how emotions, perceived threat, and prejudice come together to influence attitudes to help asylum seekers. Furthermore, it is still unclear whether Europeans distinguish between European and other asylum seekers. Therefore this study asked Europeans about their perception of asylum seekers form Ukraine, Syria, and Somalia. It examined possible differences in emotions, perceived threat, prejudice, and attitudes to help people from each of these countries. Furthermore, the theoretical understanding of the relationship between these variables was statistically examined, to explore how they fit together and inform each other." (p. 3)

"There is empirical evidence that perceived threat, emotions, and prejudice have an effect on attitudes to help refugees or asylum seekers. However their inter-relationship, particularly how perceived threat, emotions, and prejudice come together to impact attitudes to help, has not previously been examined. In addition, while there are indications that European participants would look more favourably upon Ukrainian asylum seekers, as compared to Middle Eastern or African ones, empirical evidence is currently lacking." (p. 7-8).

- These claims seem exaggerated, even considering the literature that is referenced in this paper. I believe we have sufficient knowledge to suggest specific hypotheses and mechanisms. In light of this, I have provided additional literature to further support these concepts.

The findings from the Two-way ANOVA should be reported, showing the effects of the country (UK vs. Malta), groups (Ukrainians vs. Syrians vs. Somalis), and their interactions.

In Table 4, it is noted that the values differ across two diagonals. To simplify, it would be best to retain only the values below the diagonal.

I share my common concerns about mediation analyses. I would be more careful in using the terms of mediation in the Results section giving priority to the terms of an indirect effect because non-longitudinal data make problematic a claim about mediation (see e.g., O’Laughlin, Martin, & Ferrer, 2018). Moreover, multiplying across paths to get total path strengths that they are statistically significant means nothing, as the 2019 special issue of The American Statistician makes clear and as it was explained a long time ago in the 2015 Basic and Applied Social Psychology editorial (and see Kline, 2015). I believe that a better way would be to manipulate variables that you think mediate. I will explain now. Suppose you think B mediates between A and C. Well, then, suppose you perform a manipulation to fix B at a particular level. One way is to drive B to a floor or ceiling but this is not the only way. Well, then, when B is fixed, manipulating A should no longer cause C; whereas when B is not fixed, manipulating A should cause C. This would be a much more convincing demonstration of the mediation than a mediation analysis, which is full of flaws. The only drawback is that such a design is not always possible. Therefore, I propose to add to the limitation section some of these concerns.

Sociostructural variables may be more important than the threat (Grigoryev et al., 2019; Savaş et al., 2021). However, the current body of research predominantly emphasizes intergroup relations within nations, often neglecting the disparity in status between countries due to global inequality. This is evident when considering how ethnic groups from countries that are closer to the equator - generally warmer climates and typically less wealthy - are frequently assigned lower status evaluations; such groups are also often subject to stereotypes of lesser competence and warmth (Grigoryev, 2022). It is important to recognize that darker-skinned individuals are possibly panculturally judged more negatively, given that skin color often serves as an indication of geographical proximity to the equator. Therefore, it appears that the socio-ecological explanation carries more weight than the social structural explanation in this context.

As another limitation, it would be useful to note that not only all immigrant groups are perceived equally (e.g., Schmidt, 2021), but it is important to consider benefits in addition to threats (Tartakovsky & Walsh, 2020).

Lastly, the paper requires further proofreading for improved English.

References

Aubé, B., & Ric, F. (2019). The Sociofunctional Model of Prejudice: Questioning the Role of Emotions in the Threat-Behavior Link. International Review of Social Psychology, 32(1), 1. https://doi.org/10.5334/irsp.169

Cuddy, A. J. C., Fiske, S. T., & Glick, P. (2007). The BIAS map: Behaviors from intergroup affect and stereotypes. Journal of Personality and Social Psychology, 92(4), 631–648. https://doi.org/10.1037/0022-3514.92.4.631

Hainmueller, J., & Hopkins, D. J. (2015). The hidden American immigration consensus: A conjoint analysis of attitudes toward immigrants. American Journal of Political Science, 59(3), 529–548. https://doi.org/10.1111/ajps.12138

Grigoryev, D. (2022). Ethnic stereotype content beyond intergroup relations within societies: Exploring the North-South hypothesis for competence and warmth. Cross-Cultural Research, 56(4), 345–384. https://doi.org/10.1177/10693971221080618

Grigoryev, D., Fiske, S. T., & Batkhina, A. (2019). Mapping ethnic stereotypes and their antecedents in Russia: The Stereotype Content Model. Frontiers in Psychology, 10(1643), 1–21. https://doi.org/10.3389/fpsyg.2019.01643

Kauff, M., Asbrock, F., Wagner, U., Pettigrew, T. F., Hewstone, M., Schäfer, S. J., & Christ, O. (2017). (Bad) Feelings about Meeting Them? Episodic and Chronic Intergroup Emotions Associated with Positive and Negative Intergroup Contact As Predictors of Intergroup Behavior. Frontiers in Psychology, 8. https://doi.org/10.3389/fpsyg.2017.01449

Kline, R. B. (2015). The mediation myth. Basic and Applied Social Psychology, 37(4), 202–213. https://doi.org/10.1080/01973533.2015.1049349

Kotzur, P. F., Schäfer, S. J., & Wagner, U. (2019). Meeting a nice asylum seeker: Intergroup contact changes stereotype content perceptions and associated emotional prejudices, and encourages solidarity‐based collective action intentions. British Journal of Social Psychology, 58(3), 668–690. https://doi.org/10.1111/bjso.12304

Lee T. L., & Fiske S. T. (2006). Not an outgroup, not yet an ingroup: Immigrants in the Stereotype Content Model. International Journal of Intercultural Relations, 30(6), 751–768. https://doi.org/10.1016/j.ijintrel.2006.06.005

Meuleman, B., Abts, K., Slootmaeckers, K., & Meeusen, C. (2019). Differentiated Threat and the Genesis of Prejudice: Group-Specific Antecedents of Homonegativity, Islamophobia, Anti-Semitism, and Anti-Immigrant Attitudes. Social Problems, 66(2), 222–244. https://doi.org/10.1093/socpro/spy002

O’Laughlin, K. D., Martin, M. J., & Ferrer, E. (2018). Cross-sectional analysis of longitudinal mediation processes. Multivariate Behavioral Research, 53(3), 375–402. https://doi.org/10.1080/00273171.2018.1454822

Savaş Ö., Greenwood R. M., Blankenship B. T., Stewart A. J., & Deaux K. (2021). All immigrants are not alike: Intersectionality matters in views of immigrant groups. Journal of Social and Political Psychology, 9(1), 86–104. https://doi.org/10.5964/jspp.5575

Schmidt, K. (2021). The dynamics of attitudes toward immigrants: Cohort analyses for Western EU member states. International Journal of Comparative Sociology, 62(4), 281–310. https://doi.org/10.1177/00207152211052582

Seger, C. R., Banerji, I., Park, S. H., Smith, E. R., & Mackie, D. M. (2017). Specific emotions as mediators of the effect of intergroup contact on prejudice: Findings across multiple participant and target groups. Cognition and Emotion, 31(5), 923–936. https://doi.org/10.1080/02699931.2016.1182893

Sevillano V., & Fiske S. T. (2013). Ambivalence toward immigrants: Invaders or allies? In Grigorenko E. L. (Ed.), US Immigration and Education: Cultural and Policy Issues across the Lifespan (pp. 97–118). Springer Publishing Company.

Tartakovsky, E., & Walsh, S. D. (2020). Are some immigrants more equal than others? Applying a Threat-Benefit Model to understanding the appraisal of different immigrant groups by the local population. Journal of Ethnic and Migration Studies, 46(19), 3955–3973. https://doi.org/10.1080/1369183X.2019.1565402

Witkower, Z., Mercadante, E. J., & Tracy, J. L. (2020). How affect shapes status: Distinct emotional experiences and expressions facilitate social hierarchy navigation. Current Opinion in Psychology, 33, 18–22. https://doi.org/10.1016/j.copsyc.2019.06.006

Reviewer #2: The paper "Emotions, Perceived Threat, Prejudice, and Attitudes to Help Ukrainian, Syrian, and Somali Asylum Seekers" is presented as a timely and interesting survey that offers a new perspective to investigate the link between emotions, perceived threat, and prejudice. The research seems sounding and the manuscript is generally well-written (although I would recommend re-reading by a native English speaker).

The authors report that European participants show Europe show higher warmth and attitudes to help, and lower contempt, classical prejudice, and conditional prejudice in relation to Ukrainian versus Syrian or Somali asylum seekers. Finally, authors report that prejudice mediated the relationship between emotions (contempt and warmth) and perceived threat and attitudes to help.

I think the article could be considered for publication after minor revisions.

- Introduction section: I suggest to add some lines and other references to literature to better explain the relationship between stereotype and prejudice

- Also, final discussion should be more tied to theoretical constructs presented in the introduction

- In terms of presenting the results, the authors are advised to remove the specular values below the diagonal in Table 4.

6. PLOS authors have the option to publish the peer review history of their article (what does this mean?). If published, this will include your full peer review and any attached files.

Reviewer #1: No

Reviewer #2: No

---

## [Author Response · Author response to Decision Letter 0]

21 Jul 2023

Journal requirements

1. Style : Changed, in line with the templates sent to me. 

2. Additional details re participant consent: Participants marked their consent on the online survey, and those who selected ‘no’ to any of the consenting questions were unable to complete the survey, and were simply thanked for their time.

3. Data availability: Data set uploaded

Reviewers’ comments:

1. Writing style: The paper has been reviewed by a very experienced researcher colleague, for whom English is a first language. Changes to aid readability have been made throughout, with the reordering of some arguments to enhance clarity.

2. Prejudice and stereotypes: Inserted more literature about prejudice, and highlighted the link to stereotypes.

Stereotypes are underlying beliefs about a group of people, and attributions of specific characteristic to them; they can be positive or negative [44,45]. Some research has found that stereotypes about immigrants tend to be ambivalent, for example low warmth and high competence, or high warmth and low competence [46]. Refugees as a category are generally perceived to be low on both warmth and competence, though the background of the refugees has some impact on perception of warmth [47]. Stereotypes can lead to prejudice, and are therefore key to our understanding of this phenomenon.

Prejudice tends to be a devaluation of people because of their group membership, leading to potential harm or negative consequences [48]. It relates to a judgement made before, or irrespective of, actually meeting the person. For instance, dark-skinned young people in Australia have been perceived as religiously-indoctrinated terrorists simply because of their appearance [44,49]. Prejudice is a set of hostile attitudes or feelings towards someone [44], following an evaluation of the group the person belongs to [45]. It is both irrational, in that prejudice persists in the face of evidence to the contrary, and negative [44]. Prejudice can be seen as an evaluation that includes both cognitive and affective responses [29], and often reflects a perception of the other person as low on warmth and competence [50].

3. The use of “Warmth” as a label for one of the factors is confusing in relation to Warmth and Competence as identified by Fiske: Relabelled the factors Warmth and Contempt to Positive Emotions and Negative Emotions.

4. “These claims seem exaggerated, even considering the literature that is referenced in this paper”, expressed with regard to material in page 3 and pages 7-8: Reviewed further literature, focussing on material suggested by Reviewer 1, and integrated this material into the paper. 

What was previously page 3: There is agreement in the current research literature that perceived threat influences cognitive evaluations, such as prejudice, and negative emotions towards people [15,16]. In addition, empirical studies have demonstrated that prejudice and negative emotions increase restrictive attitudes towards refugees and reluctance to help them [17,18]. However there is a gap in the literature concerning how emotions, perceived threat, and prejudice may come together to influence attitudes towards helping different groups of asylum seekers. Research has shown that refugees, asylum seekers, undocumented migrants or other incomers who are Muslim or not White are seen as a drain on a country, in comparison to generally-White migrants (such as Germans, Canadians or British) who are perceived as assets [19]. However, it is still unclear whether European citizens distinguish between asylum seekers from Europe and those from other parts of the world. This paper presents a study of Europeans’ perceptions of asylum seekers from Ukraine, Syria, and Somalia. The research investigated possible differences in the negative emotions, perceived threat, prejudice and attitudes that Europeans hold towards people arriving from each of these countries. In addition, the theoretical understanding of the relationship between these variables was statistically examined, to explore how they fit together and inform each other.

What was previously pages 7-8: There is empirical evidence that perceived threat, emotions, and prejudice have an effect on attitudes towards helping refugees or asylum seekers. The inter-relationship between these factors has been measured in relation to outgroups. For example, perceived threat predicts emotions and behavioural intentions, and emotions predict behaviour intentions [61]. However participants in that study were not asked to consider asylum seekers. In addition, it is argued that different outgroups are perceived as posing different types of threat, also depending on the ingroups’ context [62]. While there are indications that European participants would look more favourably upon Ukrainian asylum seekers, as compared to Middle Eastern or African ones, empirical evidence is currently lacking.

5. Table 4: Kept only the values above the diagonal (Reviewer 2’s suggestion).

6. 2-way ANOVA: Results are now reported

In terms of participants’ country (see Table 2), there were significant main effects for negative emotions (F(1,281)=6.05, p<.05, ηp2=.02) and perceived threat (F(1,281)=4.60, p<.05, ηp2=.02). That is, participants in Malta reported higher ratings on negative emotions and conditional prejudice, and lower ratings on positive emotions and attitudes towards helping, than participants in the UK. With regard to asylum seekers’ country of origin, there were significant main effects for negative emotions (F(2,281)=14.05, p<.001, ηp2=.09), positive emotions (F(2,281)=20.39, p<.001, ηp2=.13), conditional prejudice (F(2,281)=3.87, p<.05, ηp2=.03), and attitudes towards helping (F(2,281)=3.295, p<.05, ηp2=.02). None of the interaction effects were significant.

7. Mediation analysis: Reviewer 1 asked to add to the limitations section some of the concerns about this type of analysis. Done – see below.

Finally, in terms of the mediation analysis, evidencing mediation is challenging [73], and the mediation analysis in this study has its weaknesses. It is acknowledged that caution should be used when inferring mediation from cross-sectional data, as this type of data collection does not account for the temporal element of the experience of variables [74]. That is, a cross-section mediation analysis may yield different results when compared to a longitudinal study examining the same variables. 

 In the Results section, I now generally refer to indirect effects, rather than mediation. There are two references to mediating variables, simply because it was difficult to reword ‘mediating variable, in a mediation analysis, without the language becoming cumbersome, e.g. Hence negative emotions was a significant mediating variable in these relationships, with the indirect effects being larger than the direct effects. There are also three references to prejudice as a mediating variable in the Discussion, for this same reason. If the editor still believes that this should be amended, I will make sure to do this. The wording in the Abstract has been amended to avoid ‘mediating variable’.

8. Acknowledging some of the sociostructural variables: This has now been acknowledged. 

In addition, there is evidence that ethnic groups from lower latitudes (closer to the equator) are perceived as less trustworthy, competent, and capable, when compared to people from higher latitudes [37]

9. Tying final discussion more closely to the theoretical concepts presented in the introduction: This has been done, and the below is some of the new material, which is spread across various sections of the Discussion. 

The literature about intergroup threat identifies two types of threat: realistic and symbolic [25]. The confirmatory factor analysis in the current study questions this distinction. Looking at media and political discourses, politicians may emphasise symbolic threat, with media focussing more on realistic threats [71]. Possibly the various discourses lead people to experience threat more ‘holistically’, rather than in its theoretical subtypes. Further study could examine how people experience these theoretical sub-types of threat. 

The finding that both emotions and perceived threat have an effect on prejudice confirms the understanding of prejudice as having both an affective and a cognitive element [29].

Recognising that people evaluate asylum seekers as more or less deserving of help [57], we can now understand further how these assessments are made. While perceived threat does influence intentions to help [60], and is associated with more negative attitudes [15,31], this involves a further evaluation of asylum seekers. Hence there is a two-step evaluation: level of threat, and negative attitudes (prejudice).

Both positive and negative emotions had a stronger relationship with classical prejudice than with conditional prejudice. With classical prejudice being more overt or blatant [54,55], this suggests that conditional prejudice is possibly not simply a more acceptable way to articulate prejudicial evaluations, but perhaps a less strong or less negative bias. That is, the distinction might be more of experience than of expression. 

The indirect effect of perceived threat on attitudes towards helping (via prejudice) was stronger than that of emotions on attitudes towards helping. This indicates a stronger cognitive than emotional, effect on attitudes towards helping. In other words, what people think about asylum seekers and possibly how they perceive them, and the implications of accepting them, has a stronger effect than their feelings about asylum seekers. However, unsurprisingly, it may be difficult to separate the different influences. There is agreement in the current research literature that perceived threat influences negative attitudes and emotions towards refugees [15,16]. Perhaps because of indications that cognition and emotions interrelate [72], it is necessary to focus on how they work together to influence attitudes towards asylum seekers. This is an outstanding point of interest which requires further investigation. 

10. Considering benefits of asylum seekers: This has been added as a limitation/area for further research. 

As recent studies are doing [28], it would also be helpful to explore how people from host countries think refugees may bring benefits, delving into potential impact of asylum seekers’ country of origin with regard to this.

---

## [Editor Report · Decision Letter 1]

4 Aug 2023

Emotions, Perceived Threat, Prejudice, and Attitudes towards Helping Ukrainian, Syrian, and Somali Asylum Seekers

PONE-D-23-14284R1

Dear Dr. Xuereb,

We’re pleased to inform you that your manuscript has been judged scientifically suitable for publication and will be formally accepted for publication once it meets all outstanding technical requirements.

Kind regards,

Stefano Triberti, Ph.D.

Academic Editor

PLOS ONE
---

## [Editor Report · Acceptance letter]

10 Aug 2023

PONE-D-23-14284R1 

Emotions, perceived threat, prejudice, and attitudes towards helping Ukrainian, Syrian, and Somali asylum seekers 

Dear Dr. Xuereb:

I'm pleased to inform you that your manuscript has been deemed suitable for publication in PLOS ONE. Congratulations! Your manuscript is now with our production department. 

Kind regards, 

on behalf of

Prof. Stefano Triberti 

Academic Editor

PLOS ONE